# The Interplay between Telomeres, Mitochondria, and Chronic Stress Exposure in the Aging Egg

**DOI:** 10.3390/cells11162612

**Published:** 2022-08-22

**Authors:** Aksinya Derevyanko, Agnieszka Skowronska, Mariusz T. Skowronski, Paweł Kordowitzki

**Affiliations:** 1Molecular Neurobiology Laboratory, Salk Institute for Biological Studies, La Jolla, CA 92037, USA; 2Department of Human Physiology and Pathophysiology, Collegium Medicum, University of Warmia and Mazury, 10-719 Olsztyn, Poland; 3Department of Basic and Preclinical Sciences, Faculty of Biological and Veterinary Sciences, Nicolaus Copernicus University, 87-100 Torun, Poland

**Keywords:** oocyte, egg, aging, telomeres, mitochondria, mitochondrial function, telomere shortening, oxidative stress, ROS

## Abstract

While at the organismal level, biological aging can be estimated by telomere length and DNA methylation signatures, reliable biomarkers that can predict reproductive age are much needed to gauge the quality of an oocyte. Reproductive medicine and fertility centers often merely quantitate the ovarian reserve to predict the likelihood of fertilization and pregnancy in women of advanced reproductive age. It is highly important to address the level of age-related decline in oocyte quality since it leads to an increased risk of miscarriages and aneuploidy. Conversely, the pathways behind oocyte aging remain, in large part, elusive. Telomere shortening upon chronic stress exposure regulates mitochondria function and biogenesis by various pathways; therefore, establishing a link between these two important players and extrapolating them for the aging of oocytes will be the purpose of our commentary.

## 1. Introduction

The burden of reproductive senescence in women on public health is of high importance. In present-day society, the problem of the age-related decline in female fertility is highly relevant since the age of first pregnancy has been progressively increasing. This is compounded by increased mean life expectancy that is not matched by women’s reproductive lifespan.

Noteworthy, chronic stress is a central player impacting human health and aging and is linked to the development of many diseases [1,2]. With regards to oocyte aging, a better understanding of the pathways by which chronic stress affects the oocyte developmental competence in women of advanced age is important to counteract this process and to develop novel therapeutic strategies [2].

There are two main hypotheses describing the age-related decline in oocyte quality. The first hypothesis is related to the physiological selection process of follicular development, meaning that the highest-quality oocytes are released from the follicle at a very young reproductive age. Consequently, those oocytes with diminished quality are ovulated at a more advanced age, and, therefore, these oocytes are exposed much longer to chronic stress (Figure 1). Interestingly, in a recent study in mice, researchers found that a reduction in the number of ovulations during life minimizes the risk of oocyte aneuploidy in advanced-age mothers [3]. The second hypothesis is related to aging itself, affecting the oocytes that remain arrested in prophase I before being selected for ovulation [4].

## 2. The Link between Mitochondrial Dysfunction and Telomeres

A key role in fertility and oocyte quality impairment with aging has been attributed to mitochondrial dysfunction [4,5,6,7]. Mitochondria play a unique role in the oocyte since they provide energy to support transcription and protein synthesis during oocyte maturation and fertilization [7]. Therefore, mitochondrial quality is crucial for the oocyte condition and developmental competence [7]. Chromosome segregation errors, such as trisomy of chromosome 23, have been reported to increase as women age [7]. Interestingly, mitochondrial dysfunction is one of the key factors responsible for chromosomal segregation errors during meiosis of mammalian oocytes [7]. Further, mitochondrial dysfunction was classified as one of the major hallmarks of aging [8]. Mitochondria and mtDNA are highly abundant in mature human oocytes [9], which is necessary to provide enough energy in the form of ATP to progress from fertilization to the blastocyst [10,11]. Indeed, the fate of the embryo is highly dependent on oocyte mitochondria quality since, after fertilization, the paternal mitochondria are degraded [7,11]. Preovulatory age-related mitochondrial defects are often associated with changes in mtDNA copy number as well as in different mtDNA defects such as mutations and deletions [7,11]. Damages in mtDNA can occur upon elevated levels of reactive oxygen species (ROS) due to oxidant/antioxidant imbalance. Age-related ATP deficiency, insufficient production of coenzyme Q (CoQ), lower inner membrane potential, as well as a decline in mitochondria number were all shown to contribute to oocyte aging [7,11]. Oxidative stress is a very strong biological stimulus, influencing, among others, the reproductive system and especially the female gamete [12,13]. Telomere attrition was proposed to be another major contributor to oocyte aging [14]. The “end-replication problem” and lack of telomerase activity is the main cause of telomere dysfunction in proliferating somatic cells [15]. More precisely, during cell division, there is an incomplete replication at the end of linear DNA, leading to telomere shortening. In nondividing cells, such as oocytes, telomere damage is attributed to oxidative DNA damage of guanine-rich telomeric repeats upon exposure to ROS, as well as to epigenetic and environmental factors, diet, or lifestyle [16,17]. In somatic cells, activation of the DNA damage signaling cascade at shortened or damaged telomeres eventually leads to the activation of apoptotic or/and cellular senescence programs [18]. To what extent this process is triggered in oocytes requires further elucidation. However, it has been reported that genetically induced telomere attrition in mice phenocopies reproductive aging in women [16,19].

The first direct link between telomere and mitochondrial dysfunction, which could be of relevance for the mammalian oocyte, was shown to be mediated via the P53-peroxisome proliferator-activated receptor-γ coactivator alpha/beta (PGC1a/b) pathway (Figure 2). DePinho and colleagues [20] observed that telomerase deficiency leads to repression of mitochondria regulators PGC1a and PGC1b, leading to a decrease in the number of mitochondria and mtDNA. This process is mediated by p53 since the loss of p53 restored PGC expression and mitochondria homeostasis [20]. Other pathways linking telomere shortening and mitochondria malfunction are related to SIRT1 activity [2]. The NAD-SIRT1-PGC-1a axis is initiated by DNA damage signaling such as the one at dysfunctional telomeres. DNA repair causes the consumption of NAD^+^ and, therefore, the loss of NAD^+^-dependent deacetylase sirtuin 1 (SIRT1) activity, which affects mitochondrial health via the PGC-1a [21] or PGC-independent pathway [22]. In addition, DNA repair at telomeres can lead to activation of the mTOR cascade and PGC-1b-dependent mitochondria biogenesis (Figure 2). Mitochondria imbalance leads to ROS-mediated DNA damage response (DDR) [23], creating a feedback loop on telomeres and persistent DNA damage signaling [2]. It is worth mentioning that the presence of the shelterin complex at telomeres protects from the generation of the DNA damage response (DDR) [2].

Another important link between telomeres and mitochondria is attributed to cortisol levels. On the one hand, as shown in Figure 1, cortisol is able to provoke increased metabolic rates and mitochondrial activity, which, in consequence, leads to an elevated level of ROS [24]. On the other hand, cortisol might impact the oxidative balance directly via either genomic [25] or nongenomic pathways [26]. It is well known that telomeres are particularly vulnerable to oxidative damage [27]. Interestingly, it was revealed in a recent review that there is a negative correlation between markers for oxidative stress and telomere length [28]. In line with this, ROS has been shown to induce single-strand breaks (SSBs) at telomeres or could be responsible for replication fork collapse and telomere loss [29].

Being that telomere biology and mitochondrial function play an important role in oocyte quality, these parameters were suggested as markers for reproductive potential. More research was conducted into telomere assessment; however, few studies also addressed mtDNA. Several studies identified a positive correlation between ovarian function and mtDNA content along with telomere metabolism in follicular cells. Higher mtDNA content in cumulus cells resulted in better-quality embryos in IVF procedures [30,31]. Similar results were obtained for cumulus cells with longer telomeres [32]. Furthermore, telomeres appeared shorter in granulosa cells from women with premature ovarian insufficiency (POI) compared with healthy controls [33,34]. Telomerase activity (TA) was also lower in women with POI [33,34]. Likewise, TA appeared to be a better predictor than telomere length for pregnancy outcomes in women undergoing IVF who had normal ovarian function [35]. Due to the difficulty in sampling follicular cells, peripheral blood leukocytes (PBLs) were proposed to be used as an alternative cell type, which is more suitable for routine measurements. A number of publications identified a positive association of ovarian health with mtDNA [36,37] and telomere function [34,38] in peripheral blood cells. To date, however, the results are still controversial [39]. A recent study could not translate data on telomere length in leukocytes to follicular cells, highlighting different mechanisms for telomere maintenance in ovary and peripheral tissues [40]. Furthermore, another study could not confirm the correlation between telomere or mtDNA content and IVF success [41]. This discrepancy in results could partially arise from using different populations in terms of age and ovarian health status. Nonetheless, thus far, this controversy in data makes it challenging and requires more evidence to translate the assessment of telomere length and mtDNA as ovarian reserve biomarkers to clinics.

## 3. Conclusions

Mitochondria are crucial powerhouses in oocytes. With advancing maternal age, a concomitant decline in mitochondrial number and quality in oocytes is observed, indicating that mitochondrial function appears to be a key determinant of oocyte and embryo developmental competence. Excessive and chronic exposure to oxidative stress and reactive oxygen species, a byproduct of mitochondrial metabolism, is harmful to the chromosomes of the egg, specifically for guanine-rich telomeric hexamers. This link between mitochondria and telomeres sparked interest within the reproductive field, but more research, especially into oocytes, is required to better understand the relevance of these two key hallmarks of aging for the oocyte. The identification of accessible and noninvasive biomarkers, such as potentially telomere and mitochondria markers, would be especially helpful in determining which therapeutic strategy is adequate for individual patients. Taken together, the purpose of this commentary was to stimulate new research to elucidate the molecular mechanisms between telomeres and mitochondria during the aging process of oocytes. Therefore, novel investigations can contribute to developing new strategies to enhance and prolong the reproductive life span.

## Figures and Tables

**Figure 1 cells-11-02612-f001:**
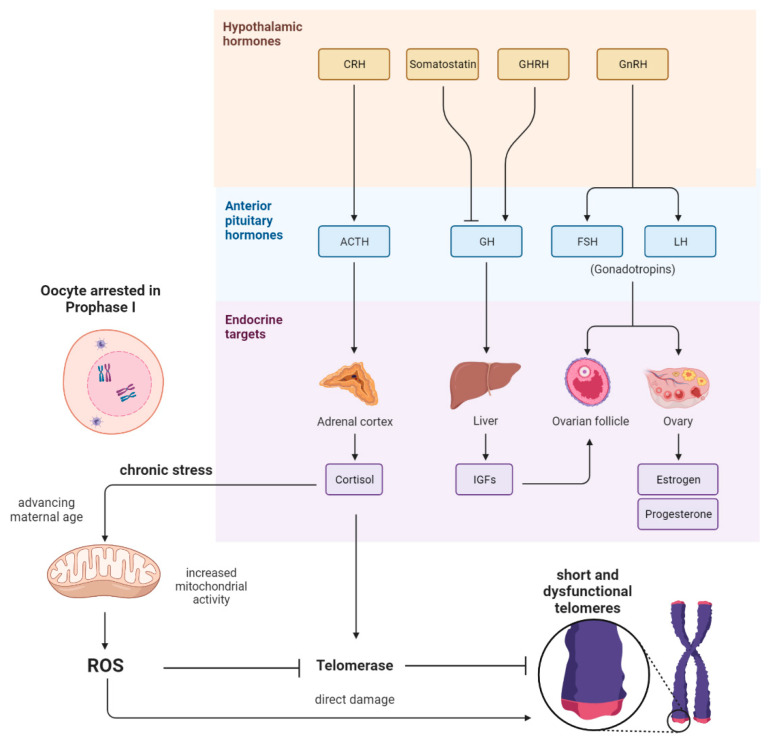
Scheme showing the pathways and interplay between the endocrine system and the reproductive system and their potential influence on the oocyte. After the hypothalamic CRH and ACTH from the pituitary gland are released, cortisol is secreted by the adrenal cortex. In consequence, there is elevated mitochondrial activity in somatic cells; therefore, reactive oxygen species (ROS) are generated. High levels of ROS could cause the oxidation of guanin-rich telomeres in women’s oocytes. Cortisol might also affect telomerase activity directly. Further, IGF1, FSH, and LH do also influence the development and maturation of ovarian follicles, which indirectly appear to influence the aging of oocytes when ovulations are altered.

**Figure 2 cells-11-02612-f002:**
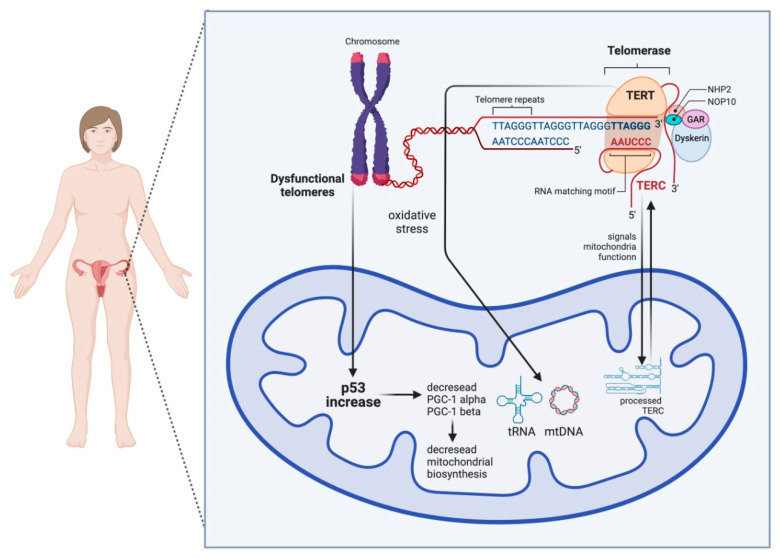
Scheme showing the bilateral crosstalk between telomeres and mitochondria in the ooplasm. On the one hand, telomere damage lowers mitochondrial biogenesis via the activation of p53, which, in consequence, negatively influences the PGC-1α and PGC-1β promoters. On the other hand, mitochondrial dysfunction can result in short and dysfunctional telomeres. TERT stands for telomerase reverse transcriptase and works as a catalytic subunit of the enzyme telomerase, and TERC stands for telomerase RNA component.

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
