# Peer review of "The Interplay between Telomeres, Mitochondria, and Chronic Stress Exposure in the Aging Egg"

_cells, 2022, doi:10.3390/cells11162612_

Round 1

Reviewer 1 Report

The presentation of facts within this commentary suggests that they have been proven in the oocyte, for most of statements this is not the case and it is not clearly delineated that these findings are in other cell types or what cells/tissues those are.  Nor do the authors really acknowledge the uniqueness of the oocyte as it relates to mitochondrial biology and that what is occurring in all of these other cells is occurring in transcriptionally active cells,  this not the case in the transcriptionally quiescent oocyte. 

Line 29.  Statement ‘is still up to date”.  Confusing not clear what the authors intention was.  Line 30.  The cause of psychological stress was too removed from Women’s fear to make sense.  Remainder of sentence is not clear either. 

Line 33. I would agree with the sentiment of this statement. However, before one can counteract, one must actually first prove that psychosocial stress actually has any effect on the oocyte. 

Line 44  “,” usage confounds the meaning of sentence.

Line 49  “on” in?

Line 53 – Too many ideas in one sentence.  

Line 59 “Besides the before mentioned psychological stress” I have a problem with this statement,  as no documentation of psychological  impacting oocytes was ever provided.   The latter half of sentence is ok.

The “end-replication problem” should be defined.

Line 84 – Many statements are not oocyte specific and this should be noted.

Line 96 – I would like to see a reference for this statement. 

Line 97 – Cortisol before CRH, ACTH that is not what the figure shows or makes sense with what is commonly understood with the HPA axis.  

Author Response

MS ID: cells- 1812761

Dear Academic-Editors and Reviewers,

Thank you for inviting us to respond to the very thoughtful and constructive reviewer comments. We greatly appreciate the reviewers time and believe our revised manuscript has become more well-rounded as a result.

We have incorporated all suggestions throughout the manuscript, and changes are highlighted in red. Below is a point-by-point response to reviewers’ and Editor’s comments to clarify which edits were made.

We are happy to respond to additional requests if they arise.

Sincerely,

Paweł Kordowitzki

Please note our following explanations:

Detailed answers to reviewer 1

REVIEWER: “The presentation of facts within this commentary suggests that they have been proven in the oocyte, for most of statements this is not the case and it is not clearly delineated that these findings are in other cell types or what cells/tissues those are.  Nor do the authors really acknowledge the uniqueness of the oocyte as it relates to mitochondrial biology and that what is occurring in all of these other cells is occurring in transcriptionally active cells,  this not the case in the transcriptionally quiescent oocyte.”

ANSWER: We want to thank the reviewer for his/her thorough review of our manuscript.  

In response, we have re-written the abstract, we clarified in the main text the uniqueness of oocytes, and to which cell types we are referring, and decided to add a paragraph on the clinical utility and a  conslusion section as follows:

Being that telomere biology and mitochondrial function play an important role in oocyte quality, these parameters were suggested as markers for reproductive potential. More research was conducted on telomere assessment, however few studies also addressed mtDNA. Several studies identified a positive correlation between ovarian function and mtDNA content along with telomere metabolism in follicular cells. Higher mtDNA content in cumulus cells resulted in better quality embryos in IVF procedure [30,31]. Similar results were obtained for cumulus cells with longer telomeres [32].  Furthermore, telomeres appeared shorter in granulosa cells from women with Premature Ovarian Insufficiency (POI) compared to healthy controls [33,34]. Telomerase activity (TA) was also lower in women with POI [33,34]. Likewise, TA appeared to be a better predictor than telomere length for pregnancy outcomes in women undergoing IVF, who had normal ovarian function [35]. Due to the difficulty of sampling follicular cells, peripheral blood leukocytes (PBLs) were proposed to be used as an alternative cell type, which is more suitable for routine measurements. A number of publications identified a positive association of ovarian health with mtDNA [36,37] and telomere function [34,38] in peripheral blood cells. To date however, results are still controversial [39]. Resent study couldn’t translate data on telomere length in leukocytes to follicular cells, highlighting different mechanisms for telomere maintenance in ovary and peripheral tissues [40]. Furthermore, another study couldn’t confirm correlation between telomere or mtDNA content and IVF success [41]. This discrepancy in results could partially arise from using different populations in terms of age and ovarian health status. Nonetheless, thus far this controversy in data makes it hard and requires more evidence to translate assessment of telomere length and mtDNA as ovarian reserve biomarker to clinics.

  1. Conclusions

Mitochondria are crucial power houses in oocytes. With advancing maternal age, a concomitant decline in mitochondrial number and quality in oocytes is observed, indicating that mitochondrial function appears to be a key determinant of oocyte and embryo developmental competence. Excessive and chronic exposure to oxidative stress and reactive oxygen species, a by-product of the mitochondrial metabolism, is harmful for the chromosomes of the egg, specifically for the guanine-rich telomeric hexamers. This link between mitochondria and telomeres sparked interest within the reproductive field but more research, especially on oocytes, is required to better understand the relevance of these two key hallmarks of aging for the oocyte.  The identification of accessible and noninvasive biomarkers, such as potentially telomere and mitochondria markers, would be especially helpful to determine which therapeutic strategy is most adequate for individual patients. Taken together, the purpose of this commentary was to stimulate new research to elucidate the molecular mechanisms between telomeres and mitochondria during the aging process of oocytes. Therefore, novel investigations can contribute to develop new strategies to enhance and prolong reproductive life span.

REVIEWER: “Line 29.  Statement ‘is still up to date”.  Confusing not clear what the authors intention was.  Line 30.  The cause of psychological stress was too removed from Women’s fear to make sense.  Remainder of sentence is not clear either. Line 33. I would agree with the sentiment of this statement. However, before one can counteract, one must actually first prove that psychosocial stress actually has any effect on the oocyte. 

 Line 44  “,” usage confounds the meaning of sentence.

Line 49  “on” in?

Line 53 – Too many ideas in one sentence.”

Line 59 “Besides the before mentioned psychological stress” I have a problem with this statement,  as no documentation of psychological  impacting oocytes was ever provided.  

Line 84 – Many statements are not oocyte specific and this should be noted.

Line 96 – I would like to see a reference for this statement.

Line 97 – Cortisol before CRH, ACTH that is not what the figure shows or makes sense with what is commonly understood with the HPA axis.

ANSWER: We thank the reviewer for his/her erudite comment, which we appreciate. In response, we have fixed all above-mentioned errors, we have improved above-mentioned statements and sentences' clarity and length throughout the MS. Moreover, we  have checked the entire MS with Grammarly and it was revised by a Native English Speaker.

REVIEWER: “The “end-replication problem” should be defined.

ANSWER: We thank the reviewer for this suggestion. In response, we have added the following:

The “end-replication problem” and lack of telomerase activity is the main cause for telomere dysfunction in proliferating somatic cells [15]. More precisely, during cell division there is an incomplete replication at the end of linear DNA, leading to telomere shortening. 

Reviewer 2 Report

Derevyanko et al discussed the possible link between telomere dysfunction, mitochondria defect and oxidative stress exposure in the aging egg process.  

Multifactorial causes of aging process have been described. Telomere dysfunction is one of important mechanisms in aging process. Telomere analysis can be added to diagnostic tests in the context of reproduction programs.  

In this commentary, the clinical utility of these tests should be discussed and the conclusion should be added. 

Author Response

MS ID: cells- 1812761

Dear Academic-Editors and Reviewers,

Thank you for inviting us to respond to the very thoughtful and constructive reviewer comments. We greatly appreciate the reviewers time and believe our revised manuscript has become more well-rounded as a result.

We have incorporated all suggestions throughout the manuscript, and changes are highlighted in red. Below is a point-by-point response to reviewers’ and Editor’s comments to clarify which edits were made.

We are happy to respond to additional requests if they arise.

Sincerely,

Paweł Kordowitzki

Please note our following explanations:

Detailed answers to reviewer 2

REVIEWER: “Telomere analysis can be added to diagnostic tests in the context of reproduction programs.  

 In this commentary, the clinical utility of these tests should be discussed and the conclusion should be added.” 

ANSWER: We want to thank the reviewer for his/her thorough review of our manuscript.  

In response, we decided to add a paragraph on the clinical utility a the conslusion as follows:

Being that telomere biology and mitochondrial function play an important role in oocyte quality, these parameters were suggested as markers for reproductive potential. More research was conducted on telomere assessment, however few studies also addressed mtDNA. Several studies identified a positive correlation between ovarian function and mtDNA content along with telomere metabolism in follicular cells. Higher mtDNA content in cumulus cells resulted in better quality embryos in IVF procedure [30,31]. Similar results were obtained for cumulus cells with longer telomeres [32].  Furthermore, telomeres appeared shorter in granulosa cells from women with Premature Ovarian Insufficiency (POI) compared to healthy controls [33,34]. Telomerase activity (TA) was also lower in women with POI [33,34]. Likewise, TA appeared to be a better predictor than telomere length for pregnancy outcomes in women undergoing IVF, who had normal ovarian function [35]. Due to the difficulty of sampling follicular cells, peripheral blood leukocytes (PBLs) were proposed to be used as an alternative cell type, which is more suitable for routine measurements. A number of publications identified a positive association of ovarian health with mtDNA [36,37] and telomere function [34,38] in peripheral blood cells. To date however, results are still controversial [39]. Resent study couldn’t translate data on telomere length in leukocytes to follicular cells, highlighting different mechanisms for telomere maintenance in ovary and peripheral tissues [40]. Furthermore, another study couldn’t confirm correlation between telomere or mtDNA content and IVF success [41]. This discrepancy in results could partially arise from using different populations in terms of age and ovarian health status. Nonetheless, thus far this controversy in data makes it hard and requires more evidence to translate assessment of telomere length and mtDNA as ovarian reserve biomarker to clinics.

  1. Conclusions

Mitochondria are crucial power houses in oocytes. With advancing maternal age, a concomitant decline in mitochondrial number and quality in oocytes is observed, indicating that mitochondrial function appears to be a key determinant of oocyte and embryo developmental competence. Excessive and chronic exposure to oxidative stress and reactive oxygen species, a by-product of the mitochondrial metabolism, is harmful for the chromosomes of the egg, specifically for the guanine-rich telomeric hexamers. This link between mitochondria and telomeres sparked interest within the reproductive field but more research, especially on oocytes, is required to better understand the relevance of these two key hallmarks of aging for the oocyte.  The identification of accessible and noninvasive biomarkers, such as potentially telomere and mitochondria markers, would be especially helpful to determine which therapeutic strategy is most adequate for individual patients. Taken together, the purpose of this commentary was to stimulate new research to elucidate the molecular mechanisms between telomeres and mitochondria during the aging process of oocytes. Therefore, novel investigations can contribute to develop new strategies to enhance and prolong reproductive life span.

Reviewer 3 Report

The article of Derevyanko et al is a mini-review, which reported some to-date knowledge about oocyte aging and the interplay between telomeres, mitochondria and stress exposure, based on 29 references.  This seems few taking in account the number of article about oocyte ageing. The review includes two high-quality figures, which resumed known pathways and molecular actors of oocyte ageing.

However, the main text is very short and very dense that makes it difficult to read. Moreover, it does not include the roles of the actors indicated in the abstract (AMH and AFC).  The abbreviations of important actors discussed in the text need to be detailed (for example, PGCs, TERT, TERC etc). Figures should be more largely explained in the text.  

English should be corrected to avoid very long sentences.

Author Response

MS ID: cells- 1812761

Dear Academic-Editors and Reviewers,

Thank you for inviting us to respond to the very thoughtful and constructive reviewer comments. We greatly appreciate the reviewers time and believe our revised manuscript has become more well-rounded as a result.

We have incorporated all suggestions throughout the manuscript, and changes are highlighted in red. Below is a point-by-point response to reviewers’ and Editor’s comments to clarify which edits were made.

We are happy to respond to additional requests if they arise.

Sincerely,

Paweł Kordowitzki

Please note our following explanations:

Detailed answers to reviewer 3

REVIEWER: The article of Derevyanko et al is a mini-review, which reported some to-date knowledge about oocyte aging and the interplay between telomeres, mitochondria and stress exposure, based on 29 references. However, the main text is very short and very dense that makes it difficult to read.

ANSWER: We want to thank the reviewer for his/her thorough review of our manuscript.  Since our article is a Commentary we decided to keep this piece short but we  decided also to add more references, and now the article contains 41 references, and we improved and extended the main text accordingly.

REVIEWER: “Moreover, it does not include the roles of the actors indicated in the abstract (AMH and AFC). The abbreviations of important actors discussed in the text need to be detailed (for example, PGCs, TERT, TERC etc). Figures should be more largely explained in the text.”

ANSWER: We thank the reviewer for his/her erudite comment, which we appreciate. In response, we decided to re-write the abstract and the main text, and we deleted the sentence regarding AMH and AFC since it is beyond this commentary to discuss these topics.  We have introduced and explained the abbreviations in the text and in the figure legend as follows:

The first direct link between telomere and mitochondrial dysfunction, which could be of relevance for the mammalian oocyte, was shown to be mediated via the P53-peroxisome proliferator-activated receptor-γ coactivator alpha/ beta (PGC1a/b) pathway (Fig.2).

TERT stands for: Telomerase reverse transcriptase and works as a catalytic subunit of the enzyme telomerase, and TERC stands for telomerase RNA component.

REVIEWER: “English should be corrected to avoid very long sentences.”

ANSWER: We want to thank the reviewer for this suggestion.  In response, we  have checked the entire MS with Grammarly and it was revised by a Native English Speaker.